# GO-SWCNT Buckypapers as an Enhanced Technology for Water Decontamination from Lead

**DOI:** 10.3390/molecules27134044

**Published:** 2022-06-23

**Authors:** Mariafrancesca Baratta, Antonio Tursi, Manuela Curcio, Giuseppe Cirillo, Fiore Pasquale Nicoletta, Giovanni De Filpo

**Affiliations:** 1Department of Chemistry and Chemical Technologies, University of Calabria, 87036 Rende, Italy; mariafrancesca.baratta@unical.it (M.B.); antonio.tursi@unical.it (A.T.); 2Department of Pharmacy, Health and Nutritional Sciences, University of Calabria, 87036 Rende, Italy; manuela.curcio@unical.it (M.C.); giuseppe.cirillo@unical.it (G.C.)

**Keywords:** lead, heavy metals, single-walled carbon nanotubes, graphene oxide, buckypaper, adsorption

## Abstract

Water decontamination is an important challenge resulting from the incorrect disposal of heavy metal waste into the environment. Among the different available techniques (e.g., filtration, coagulation, precipitation, and ion-exchange), adsorption is considered the cheapest and most effective procedure for the removal of water pollutants. In the last years, several materials have been tested for the removal of heavy metals from water, including metal-organic frameworks (MOFs), single-walled carbon nanotubes (SWCNTs), and graphene oxide (GO). Nevertheless, their powder consistency, which makes the recovery and reuse after adsorption difficult, is the main drawback for these materials. More recently, SWCNT buckypapers (SWCNT BPs) have been proposed as self-standing porous membranes for filtration and adsorption processes. In this paper, the adsorption capacity and selectivity of Pb^2+^ (both from neat solutions and in the presence of other interferents) by SWCNT BPs were evaluated as a function of the increasing amount of GO used in their preparation (GO-SWCNT buckypapers). The highest adsorption capacity, 479 ± 25 mg g^−1^, achieved for GO-SWCNT buckypapers with 75 wt.% of graphene oxide confirmed the effective application of such materials for cheap and fast water decontamination from lead.

## 1. Introduction

The incorrect disposal of heavy metal waste from different industrial plants represents a serious hazard for human health and the environment because pollutants can seriously contaminate, even at low doses, freshwater resources essential for all lifeforms [1,2,3,4]. Lead is the most frequently found heavy metal in contaminated water resources and, because of its bioaccumulation, can cause several human health problems, including damage to muscles, bones, kidneys, the nervous system, and cancer [5,6,7]. The World Health Organization fixed in 0.01 mg L^−1^ (10 ppb) the lead ions provisional guideline value for drinkable water [8]. Many technologies, including bio-treatments [9], membrane filtration [10], photocatalysis [11], ion exchange membranes [12], coagulation/chemical precipitation [13], electrochemical technologies [14], and oxidation/reduction [15], have been proposed in order to move heavy metal, and lead in particular, concentrations within the suggested limits from plant effluents. Nevertheless, some drawbacks affect these processes, such as slow and poor removal efficiencies, toxic by-products, and high costs [16,17], which can be generally bypassed by adsorption methods. In fact, adsorption processes are inexpensive and easy to perform and reuse, with no by-products [18]. Several adsorbents have been proposed for the removal of heavy metals from waste-water including activated carbon [19], clays and zeolites [20,21], nanoparticles [22,23], organic resins and polymers [24], metal oxides [25], agricultural waste products [26,27], zero-valent ions [28]. More recently, other materials have been investigated as efficient adsorbents for heavy metals ions, including single and multiwalled carbon nanotubes (SWCNTs and MWCNTs) [29,30,31], graphene oxides (GOs) [32,33,34], and organic metal frameworks (MOF) [35,36] as either single components or in mixtures. Despite their enhanced adsorption properties, due to their large surface area, porosity, and the possibility of tuning their interactions with the target pollutants by easy functionalization [37], these materials lack the possibility of easy recovery and regeneration after use due to their powdery consistency. Consequently, several researchers have tried to overcome such drawbacks by assembling such materials in buckypapers (BPs) [38,39,40].

BPs are films formed by CNTs and CNT bundles self-assembled mainly by π–π and van der Waals interactions [41,42]. Due to such self-standing properties, they are also known as CNT films or CNT papers. In addition, BPs are characterized by a porous structure, low density, and interesting thermal, rheological, and electrical properties [43]. In order to enhance their properties, BPs are used to host other materials such as MOF [36,44] and GO [42,45], leading to the formation of composite materials with superior performance in different fields such as flexible sensors [45], batteries [46], smart packaging [47], artificial muscles [48], fire retardant [49], and membrane-based processes, namely desalinization and catalysis [40,50]. In the literature, few works deal with the application of composite BPs as adsorbent membranes. Recently, SWCNT BPs incorporating MOFs have been proposed as boosted adsorbents for the recovery of rare-earth elements [44] and the selective capture of lead [36] from wastewater. The presence of MOF in SWCNT BPs was found to be beneficial for the adsorption of all tested lanthanides, with the Ce^3+^ recovery reaching the value of 263.30 mg of cerium adsorbed per gram of MOF-SWCNT BP. This was related to the alcohol functionalities of threonine within MOF pores. Similarly, the lead adsorption capacity of neat BPs increased by 42% when 25 wt.% SWCNTs were substituted by MOFs. Moreover, stable BPs for the adsorption of heavy metal ions were obtained by the noncovalent interaction of oxidized CNTs with GO [51]. The presence of nanotubes stabilized the mechanical properties of BPs but reduced the adsorption capacities for all tested heavy metals. In particular, the experimental lead adsorption capacity by BPs with GO content up to 90 wt.% was lower than the experimental maximum loading capacity by neat CNT BPs (163 ± 10 mg g^−1^) due to the presence of micro- and nano-channels among entangled CNTs.

In this paper, single-walled carbon nanotube/graphene oxide buckypapers (GO-SWCNT BPs) were prepared and characterized as novel membranes for the adsorption of lead from wastewater. The adsorption capacity and selectivity towards Pb^2+^ (both from neat solutions and in the presence of other interferents) by GO-SWCNT BPs were evaluated as a function of the increasing percentage (wt.) of GO used in their preparation. Hybrid membranes showed high lead adsorption capacity and interesting removal properties towards other heavy metal ions, allowing the possibility of multielement decontamination of wastewater.

## 2. Results

SWCNT BPs and GO-SWCNT BPs were prepared according to the wet method [52,53], including the uniform dispersion of carbonaceous compounds in a surfactant solution, vacuum filtration through a PTFE porous filter (average pore size 5 μm), several washes with methanol, drying, and, finally, peeling off from the PTFE filter.

Under the used conditions reported in the Materials and Methods section, the wet method provided self-standing and flexible BP disks for GO loadings up to 75 wt.% (Figure 1) due to the π–π and van der Waals interaction forces between SWCNTs and GOs. Unfortunately, larger GO amounts gave cracked films (see 85% GO-SWCNT BP picture in Figure 1e). Consequently, successive investigations were limited to BP samples with a maximum GO loading of 75 wt.% (0, 25, 50 and 75 wt.% GO).

As shown in Figure 1f, BPs were characterized by easy detachability from porous PTFE filter and self-sustainability for all studied GO wt. percentages. BPs were flexible disks with an average thickness of around 100 ± 2 µm and an average diameter of 38 ± 1 mm.

Figure 2 shows the morphology of neat SWCNT BP and GO-SWCNT BPs with different GO content (25, 50, and 75 wt.%). SWCNT BP morphology was characterized by the presence of the typical SWCNT bundles and clusters arising from π–π and van der Waals interactions. The found porous structure is expected to give BPs a high permeability and a large contact surface area for heavy metal adsorption. The addition of an increasing amount of GO in SWCNT BPs resulted in increased GO sheets number homogenously hosted inside the membranes.

Nevertheless, the increasing the number of empty spaces around the GO sheets for BP with higher GO loadings is expected to lead to more fragile BPs with lower mechanical properties. In fact, even if all samples looked like very similar free-standing membranes, both from a macroscopic and microscopic point of view, they were found to be characterized by decreasing values of tensile strength and fracture strain, as reported in Figure 3 and Table 1.

Despite the slight reduction in the mechanical properties, all investigated BPs can be considered highly stable membranes, and could find important application in adsorption treatments for wastewater.

The increase in GO content was also found to modify the hydrophilicity of BP top surfaces. As shown in Table 1, SWCNT BPs were characterized by lower hydrophilicity (average water contact-angle of 71.3° ± 0.5°) than GO-SWCNT BPs, which showed a gradual decrease in the water contact-angle values from 62.6° ± 0.5° (25% GO-SWCNT BP) to 41.2° ± 0.5° (75% GO-SWCNT BP). Obviously, it is expected that the decrease in water contact-angle values could make GO-SWCNT BPs more effective devices for wastewater treatments than neat SWCNT BPs due to easier wettability. On the contrary, the porosity percentage found for BPs decreased for larger GO amounts due to closer packing between GO sheets.

The lead removal properties of SWCNT and GO-SWCNT membranes were evaluated through adsorption experiments. Membrane disks were immersed in 800 mL beakers with Pb(NO_3_)_2_ water solutions at different lead concentrations (1, 10, and 50 ppm). The kinetic profiles of lead recovery in the 0–72 h interval, are reported in Figure 4.

All membranes were able to recover almost 100% of the lead present in the 1 ppm solutions, as seen in Figure 4a. The residual lead concentration was always lower than 10 ppb (7.2, 5.6, 4.6, and 3.1 ppb, respectively for increasing GO amounts), which represents an acceptable limit for drinking water [8]. In particular, 75% GO-SWCNT BPs were able to remove almost all lead ions present in the 1 ppm solution within 8 h. However, slightly different recoveries (ranging from 85% to 95% for increasing GO amounts) were found when 10 ppm lead solution was used (Figure 4b), confirming the positive effect of GO substitution on the adsorption properties of hybrid membranes. For larger concentrations (50 ppm), the lead recovery varied from 42.18% (SWNT BPs) to 47.19%, 53.02%, and 59.95% for GO amounts of 25, 50, and 75 wt.%, respectively. Such results gave a lead adsorption capacity per unit of adsorbent mass of 337 ± 13, 377 ± 9, 424 ± 13, and 479 ± 25 mg g^−1^, and confirmed again the best adsorption capacity shown by 75% GO-SWCNT BPs. The obtained adsorption capacity per unit of adsorbent mass is larger than the values recently found for the lead removal by MOF powders (108 mg g^−1^) [54] and BP membranes hosting MOF (310 mg g^−1^) [36] but lower than the lead adsorption capacity by GO flakes (555 mg g^−1^) [54]. It is worth noting that, despite the slight reduction in the adsorption capacity (−15%) due to the presence of SWCNTs, 75% of GO-SWCNT BPs are self-standing films, which can be more easily used, recovered, and regenerated than any powder-like adsorbent (see *infra*). The adsorbent physicochemical properties, such as water wettability, surface area, and chemical nature, can affect the adsorption of heavy metals in addition to the operating conditions (e.g., adsorbent amount, temperature, pH and ionic strength values, adsorption time, initial concentration, and interferent presence). The presence of surface functional groups, such as hydroxyl and carboxyl, can improve wettability and heavy metal chelation. Therefore, the increased lead adsorption capacity shown by GO-SWCNT BPs with larger amounts of GO (from 42.18% for SWNT BPs to 59.95% for 75% GO-SWCNT BP) can be attributed to the beneficial presence of functional groups able to enhance membrane wettability and metal chelation and overcome the possible drawbacks due to the reduced porosity (see Table 1).

The beneficial substitution of SWCNT with GO in BPs was further confirmed through kinetic experiments. From the literature [36,38], it is known that the rate equation for Pb^2+^ capture in solutions follows the Lagergren first-order Equation (1):(1)$$\frac{dq_{t}}{dt}=k_{1}\left(q_{e}-q_{t}\right)$$
where *k*_1_ and *q_e_* are the Lagergren adsorption rate constant (min^−1^) and the lead adsorption capacity per unit of adsorbent mass (mg g^−1^) at equilibrium, respectively. The experimental concentrations measured as a function of time, *C*(*t*), were used to determine the experimental amount, *q_exp_*(*t*), (mg g^−1^) of lead adsorbed by BPs, according to the following Equation (2):(2)$$q_{exp}\left(t\right)=\frac{C_{0}-C\left(t\right)}{m}\times V$$
where *C*_0_ and *C*(*t*) are the lead concentration in the solution at time zero and *t*, respectively. *V* is the volume of lead solution, and *m* is the mass of BPs.

Experimental data were well-fitted (*R*^2^ values always larger than 0.975) by non-linear optimization method (OriginPro 2019 Software, OriginLab Corporation, Northampton, MA, USA) [55] with the following Equation (3):(3)$$q_{t}=q_{e}\left(1-e^{-k_{1}t}\right)$$
which can be easily obtained from the integration of Equation (1).

As reported in Table 2, fittings of experimental adsorptions at 1 ppm as a function of time showed an increase in the *k*_1_ value for increasing GO amounts incorporated in SWCNT BPs. In fact, the initial *k*_1_ value of 5.44 ± 0.07 × 10^−3^ min^−1^ found for neat SWCNT BP increased to 8.81 ± 0.05 × 10^−3^ min^−1^ for 75% GO-SWCNT BP (increase ≈ 62%).

The found Lagergren adsorption rate constants are of the same order and magnitude as those recently obtained for the lead removal by BP membranes hosting MOF (14.3 ± 0.8 × 10^−3^ min^−1^) [36], but lower than the lead adsorption rate constant by GO nanosheets (200 × 10^−3^ min^−1^) [56].

To validate the good lead adsorption properties by 75% GO-SWCNT membranes even in the presence of the most found background ions, further adsorption experiments were performed with 1 ppm lead solution prepared with mineral water, where both monovalent and divalent ions (Na^+^, K^+^, Mg^2+^, and Ca^2+^) were present. The initial concentration of such ions was 6.18, 3.90, 5.96, and 15.89 ppm for Na^+^, K^+^, Mg^2+^, and Ca^2+^, respectively. As it is evident from Figure 5a, both the capture performance of hybrid membranes and the selectivity towards Pb^2+^ ions were preserved.

The observed selectivity of 75% GO-SWCNT BPs toward lead was further confirmed when other divalent and potential interferent heavy metals (Co^2+^, Zn^2+^, Cd^2+^, and Hg^2+^) were present in a simulated wastewater at the concentration of 1 ppm, as shown in Figure 5b. In particular, the Co^2+^, Zn^2+^, and Cd^2+^ recovery percentage was lower than one half than the one found for lead.

The lead capture performance was still the highest when 75% GO-SWCNT membranes were soaked in a solution containing 1 ppm of other heavy metal ions easily found in wastewater, such as Fe^3+^, Al^3+^, and Cu^2+^, see Figure 5c.

It is worth noting that, under such drastic experimental conditions, the initial lead concentration of 1 ppm was lowered to the value of about 300 ppb when a 75% GO-SWCNT BP was soaked in the solution beaker, whereas the removal efficiency for Pb^2+^, Fe^3+^, Al^3+^, and Cu^2+^ was around 67%, 62%, 52%, and 46%, respectively. Even if lower than the lead adsorption capacity, the removal percentages of all other tested heavy metals by GO-SWCNT membranes were interesting for wider decontamination of wastewater from such toxic pollutants.

Stability in the adsorption performance after regeneration is an important item for every adsorbent in view of its potential industrial applications. The adsorption performance of 75% GO-SWCNT BP was measured in 1 ppm lead solution after four regeneration processes, performed by soaking the membranes in a 10% (*v*/*v*%) aqueous solution of 2-mercaptoethanol for 24 h in order to favor the release of the adsorbed lead. As shown in Figure 6, the initial recovery found after the first immersion in the lead solution was 99.69% and decreased to 93.42% during the fifth adsorption cycle performed after the fourth regeneration process, confirming the reusability, efficiency, and stability of membranes.

Such properties make GO-SWCNT BPs suitable for industrial wastewater applications, thanks also to their easy scalability as filtration units both in parallel for large-scale treatments and in series for their increased efficiency performance.

Regarding the industrial feasibility and scale-up, at this stage, it is difficult to estimate the operating costs for lead removal by the proposed technology and compare them with those of other existing technologies. According to a previously reported estimation to produce BPs (hosting MOFs at 25 wt.% for a 15 kg scale), the costs of BPs and MOFs were estimated at around 1000 €/kg and less than 4000 €/kg, respectively [36]. Considering that the GO production costs are noticeably lower than those of MOFs [57], it is possible to assess that the production of GO-SWCNT BPs is cheaper than that of MOF-SWCNT BPs or pure-MOF-based devices.

## 3. Materials and Methods

### 3.1. Buckypaper Preparation

SWCNT BPs were prepared using mixtures of commercially available single-walled carbon nanotubes, SWCNTs, with a length longer than 5 µm and an average diameter of 1.8 ± 0.1 nm, and commercially available carboxylic acid functionalized SWCNTs (COOH-SWCNTs, 1.0–3.0 atom% carboxylic acid) with average bundle lengths from 0.5 to 1.5 µm and diameters from 4 to 5 nm, as reported in their datasheets from Sigma-Aldrich, Milan, Italy. Briefly, 50 mg SWCNT and COOH-SWCNT mixtures in the ratio 2:1 by weight were dispersed in 250 mL 0.4% TRITON × 100 water solution by an ultrasonic bath (M1800H-E, Bransonic, Danbury, CT, USA) for 30 min, then filtered through poly(tetrafluoroethylene disks (PTFE, diameter = 47 mm, average pore size = 5 µm, Durapore©, Merck KGaA, Darmstadt, Germany) with a vacuum pump (pressure = −0.04 bar), washed several times with methanol and, finally dried at room temperature.

All chemicals were reagent grade and purchased from Sigma-Aldrich, Milan, Italy.

All filtration conditions including the weight percentage between SWCNTs and COOH-SWCNTs, the amount of TRITON ×100, the sonication time, the vacuum depression, the filter porosity, and material were chosen to obtain self-standing and flexible BP disks [36] (Figure 1).

Similarly, single-walled carbon nanotube/graphene oxide buckypapers, GO-SWCNT BPs, were obtained by substituting a given weight amount (25, 50, and 75 wt.%) of SWCNT-COOH-SWCNT mixture with an identical quantity of graphene oxide, (GO, 15–20 sheets, 4–10% edge-oxidized, Sigma-Aldrich, Milan, Italy) and following the procedure previously outlined for the preparation of SWCNT BPs. As shown in Figure 1, BPs were characterized by easy detachability from porous PTFE filter and self-sustainability for all studied GO wt. percentages. The obtained membranes were flexible disks with an average thickness of 100 ± 2 µm and an average diameter of 38 ± 1 mm. The substitution of SWCNT mixture with GO amounts larger than 75 wt.% gave cracked films, as reported in Figure 1e for 85% GO-SWCNT BPs, with difficult detachability and, consequently, broken membranes. Therefore, the maximum GO loading in GO-SWCNT BPs was set at 75 wt.%.

### 3.2. Characterization Procedure

The morphology of the SWCNT and GO-SWCNT membranes was investigated by a scanning electron microscope (LEO 420, Leica Microsystems, Cambridge, England, accelerating voltage of 10 kV) after their sputtering with an ultrathin gold layer.

Static contact-angle values of SWCNT and GO-SWCNT BPs were obtained with a goniometer (Nordtest, Serravalle Scrivia AL, Italy) at 25 °C. A drop (2 µL) of water was put on five different positions of the sample surface by a micro-syringe, and angle values were evaluated by drawing the tangents on both visible edges of each droplet and calculating the average value of the measurements.

The percentage porosity of BP, *P*(%), was estimated by measuring the weight of a wetting liquid (3M-FC-40, 3M Italia Srl, Pioltello, Milan, Italy) contained in the membrane pores at 25 °C, according to the following Equation (4):(4)$$P\left(\%\right)=\frac{\frac{W_{w}-W_{d}}{d_{w}}}{\frac{W_{w}-W_{d}}{d_{w}}+\frac{W_{d}}{d_{m}}}$$
where *W_w_* and *W_d_* are the weight of the wet and dry samples, and *d_w_* and *d_m_* are the density of the wetting liquid (1.855 g cm^−3^) and BP membranes, respectively.

The tensile strength and fracture strain of SWCNT and GO-SWCNT BPs were measured on rectangular strips (3 cm in length and 5 mm in width) at a strain rate of 0.1 mm min^−1^ with a Sauter TVO-S tensile tester equipped with a Sauter FH-1k digital dynamometer and AFH FAST software (Sauter GmbH, Balingen, Germany).

### 3.3. Lead Adsorption by SWCNT and GO-SWCNT BPs

Lead adsorption by SWCNT and GO-SWCNT membranes was evaluated at 25 °C in deionized water solutions of Pb(NO_3_)_2_. SWCNT and GO-SWCNT BPs were placed in beakers containing 800 mL Pb(NO_3_)_2_ solution (lead concentration of 1, 10, and 50 ppm) and stirred by an orbital shaker (PSU-10i, Biosan, Italy) at 25 °C.

Mineral water solutions were used to test BP performance in lead (1 ppm) adsorption in the presence of interferent elements commonly found in water, such as Na^+^, K^+^, Ca^2+^, and Mg^2+^. Further investigations on the effect of the interferents’ presence on lead removal were performed by measuring the lead adsorption from two different heavy metal multielement solutions, containing Pb^2+^, Co^2+^, Zn^2+^, Cd^2+^, and Hg^2+^ ions or Pb^2+^, Fe^2+^, Al^3+^, and Cu^2+^ ions. The concentration of each heavy metal was 1 ppm in both multielement solutions.

The effect of pH on lead adsorption was tested at the following values: 4.5, 6, and 8. It was found that a 75% GO-SWCNT BP was able to recover about 99.7% of lead present in an 800 mL solution at a concentration of 1 ppm when the pH value was 6. The capture capacity decreased to about 92.0% and 89.9% when pH was 4.5 and 8, respectively. Consequently, all adsorption experiments were performed at pH 6, if not differently reported.

The heavy metal ion concentrations were determined by inductively coupled plasma-mass spectrometry (ICP-MS iCAP™ TQ Thermo Fisher Scientific, Waltham, MA, USA) equipped with a Peltier cooled high purity quartz baffled cyclonic spray chamber, a concentric borosilicate glass nebulizer, a wide 2.5 mm internal diameter quartz injector, a nickel sample and two skimmer cones with 1.1 mm and 0.5 mm diameter orifices, respectively. The ICP torch was a demountable single-piece quartz torch. Samples were collected by a Thermo Scientific™ Autosampler Housing with a peristaltic pump equipped with three-stop flared PVC pump tubing. A multielement standard solution was used to calibrate the instrument using different analytical concentrations (0.5, 5, 10, 20, 50, and 100 ppb, *R*^2^ of calibration curve ≈ 1, and a limit of detection, LOD, of about 0.015 ppb). Ultrapure deionized water (18.3 MΩ cm, Arioso, Human Corporation, Seoul, Korea) was used for the preparation of the aqueous solution after filtration by a 0.45 μm filter (Millex Syringe Filter, Merck, Darmstadt, Germany). Each experiment was performed in triplicate, and the results are reported as average values ± 3 SD.

## 4. Conclusions

In the present work, the preparation and characterization of GO-SWCNT buckypapers was reported, and their efficiency as lead adsorbents was tested as a function of different amounts of graphene oxide present in the membranes. The partial substitution of SWCNTs with GO had a beneficial effect in the adsorption properties of SWCNT BPs, which were still self-standing membranes without any important loss in their mechanical properties and adsorption capacities.

The GO-SWCNT BPs exhibited great capture properties for Pb^2+^ removal from aqueous solutions even in the presence of interferent ions, which could be present in water, such as Na^+^, K^+^, Ca^2+^, and Mg^2+^, and other heavy metal ions, which could be present in wastewater such as Co^2+^, Zn^2+^, Cd^2+^, Hg^2+^, Fe^3+^, Al^3+^, and Cu^2+^. The GO-SWCNT membranes were also characterized by interesting capture properties towards all the ions present in the multielement solutions, allowing the possibility of wider decontamination of wastewater from several toxic pollutants at the same time.

For solutions with a lead concentration of 1 ppm, all GO-SWCNT BPs were able to reduce the [Pb^2+^] to values well below the limit of 10 ppb allowed by WHO for drinkable water. The lead adsorption efficiency of the 75% GO-SWCNT BPs was maintained over five adsorption/four regeneration cycles without any important reduction (less than 7%) in the removal efficiency.

In addition to the above-mentioned features, it is important to emphasize the importance of the easy scalability of the proposed GO-SWCNT BPs as filtration units both in series for an efficiency increase and in parallel for applications in large-scale decontamination plants.

## Figures and Tables

**Figure 1 molecules-27-04044-f001:**
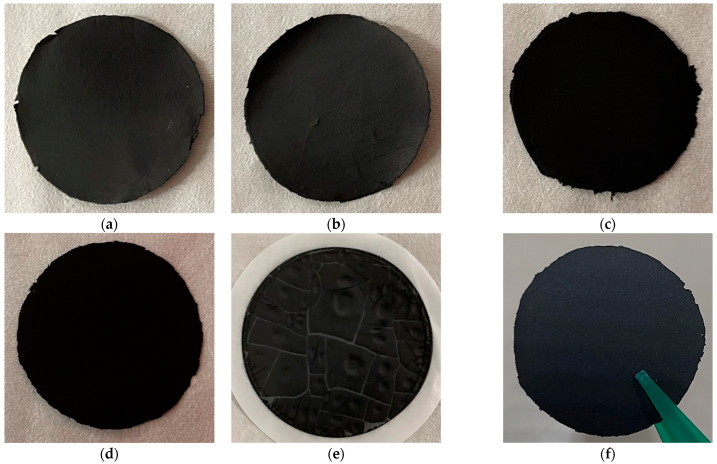
Pictures of (**a**) SWCNT BP, (**b**) 25% GO-SWCNT BP, (**c**) 50% GO-SWCNT BP, (**d**) 75% GO-SWCNT BP, and (**e**) 85% GO-SWCNT BP. All samples, except for 85% GO-SWCNT BP (which showed evident macroscopic cracks), were easy detachable from the porous PTFE filter and resulted in (**f**) self-sustainable and flexible disks.

**Figure 2 molecules-27-04044-f002:**
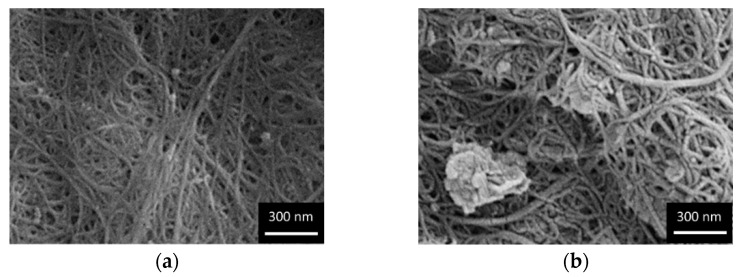
SEM images of (**a**) SWCNT BP, (**b**) 25% GO-SWCNT BP, (**c**) 50% GO-SWCNT BP, and (**d**) 75% GO-SWCNT BP, showing the presence of GO flakes homogeneously embedded in the SWCNT BP network.

**Figure 3 molecules-27-04044-f003:**
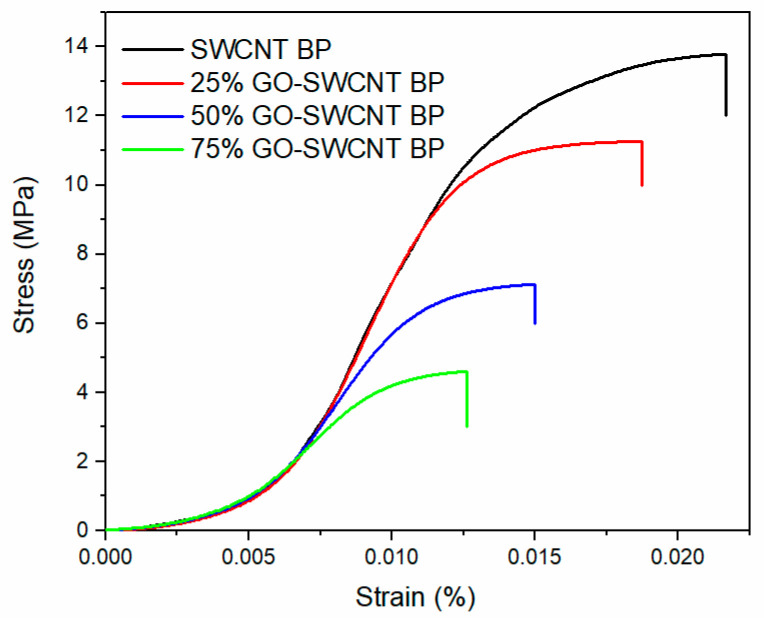
Stress-strain curves of SWCNT BP with different weight percentage of GO.

**Figure 4 molecules-27-04044-f004:**
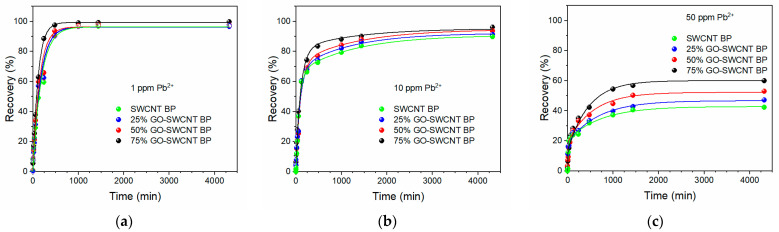
Pb^2+^ recovery by neat SWCNT BP and GO-SWCNT BPs during soaking in 800 mL of aqueous solutions with [Pb^2+^] of: (**a**) 1 ppm, (**b**) 10 ppm, and (**c**) 50 ppm.

**Figure 5 molecules-27-04044-f005:**
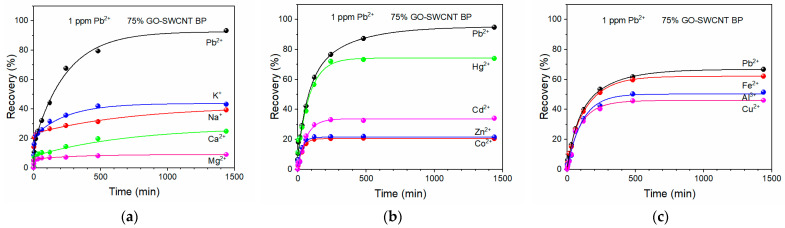
Pb^2+^ recovery by 75% GO-SWCNT BPs during the soaking in 800 mL of: (**a**) a mineral water solution with [Pb^2+^] of 1 ppm, (**b**) a multielement heavy metal solution with Pb^2+^, Co^2+^, Zn^2+^, Cd^2+^, and Hg^2+^, (**c**) a multielement heavy metal solution with Pb^2+^, Fe^3+^, Al^3+^, and Cu^2+^. The concentration of each heavy metal in the multielement solutions was 1 ppm.

**Figure 6 molecules-27-04044-f006:**
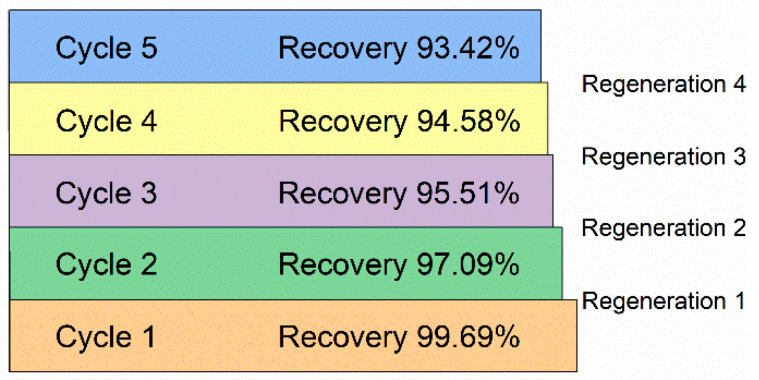
Pb^2+^ adsorption performance by 75% GO-SWCNT BPs after four regeneration processes in a 10% (*v*/*v*%) aqueous solution of 2-mercaptoethanol for 24 h. The initial lead concentration was 1 ppm.

**Table 1 molecules-27-04044-t001:** Mechanical properties, water contact-angles, and the porosity of SWCNT BP with different weight percentage of graphene oxide (GO).

Sample	Tensile Strength (MPa)	Fracture Strain (%)	Water Contact-Angle (°)	Porosity (%)
SWCNT BP	13.8 ± 0.9	2.2 ± 0.5	71.3 ± 0.5	74 ± 5
25% GO-SWCNT BP	11.3 ± 0.8	1.9 ± 05	62.6 ± 0.5	70 ± 5
50% GO-SWCNT BP	7.2 ± 0.6	1.5 ± 0.4	51.2 ± 0.5	65 ± 5
75% GO-SWCNT BP	4.7 ± 0.6	1.3 ± 0.4	41.2 ± 0.5	59 ± 5

**Table 2 molecules-27-04044-t002:** Lagergren adsorption rate constant, *k*_1_ (min^−1^), and lead adsorption capacity per unit of adsorbent mass at equilibrium, *q_e_* (mg g^−1^), for BPs with different GO loading.

Sample	*k*_1_ × 10^3^ (min^−1^)	*q_e_* (mg g^−1^)	*R* ^2^
SWCNT BP	5.44 ± 0.07	15.3 ± 0.5	0.9784
25% GO-SWCNT BP	6.07 ± 0.06	15.4 ± 0.4	0.9864
50% GO-SWCNT BP	6.88 ± 0.07	15.4 ± 0.4	0.9864
75% GO-SWCNT BP	8.81 ± 0.05	15.9 ± 0.2	0.9958

## Data Availability

Data are contained within the article.

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
