# Peer review of "GO-SWCNT Buckypapers as an Enhanced Technology for Water Decontamination from Lead"

_molecules, 2022, doi:10.3390/molecules27134044_

Round 1
Reviewer 1 Report
In this paper, original and scientifically valuable results on the adsorption capacity and selectivity of Pb2+ by SWCNT BPs were evaluated as a function of the increasing amount of GO used in their preparation (up to 75 wt.% of graphene oxide). It is shown that GO-SWCNT BPs are easily scalable, possible to recover, and efficient materials for water cleaning processes.
The manuscript is well prepared. Information in the Introduction part gives state of art analysis on heavy metals removal from water, synthesis, and use of GO-SWCNT BPs. The experimental part related to BPs synthesis and analysis, lead adsorption by SWCNT and GO-SWCNT BPs are sufficiently described. The results analysis and conclusion are in good correlation.
Despite of all benefits, the manuscript could be slightly improved:
1. It is mentioned that BPs are used to host other materials such as MOF and GO, leading to the formation of composite materials with superior performance. Would be useful to compare already existing results received by other authors with those received and described in this manuscript. It would help to confirm the novelty and added value of this research.
It is also mentioned that the use of c GO-SWCNT BPs could be a cheap and effective way for water decontamination from lead. It remains insufficiently justified in the manuscript. Comparison with other methods and materials in terms of economic price and technological efficiency should be done.
Author Response
Reply to the referees’ comments (Ms # molecules-1771507)
We thank the referees for their valuable comments and suggestions, which helped us much in the revision of the manuscript. In the following pages, we report the responses to their comments.
Reviewers' Comments to Authors
Reviewer # 1
- It is mentioned that BPs are used to host other materials such as MOF and GO, leading to the formation of composite materials with superior performance. Would be useful to compare already existing results received by other authors with those received and described in this manuscript. It would help to confirm the novelty and added value of this research.
- We agree with the comment of reviewer # 1. The sentence at lines 60-63 was integrated with the following: “in different fields such as flexible sensors [45], batteries [46], smart packaging [47], artificial muscles [48], fire retardant [49], and membrane-based processes, namely desalinization and catalysis [50, 40]. In literature, few works deal with the application of composite BPs as adsorbent membranes. Recently, SWCNT BPs incorporating MOFs have been proposed as boosted adsorbents for the recovery of rare-earth elements [44] and the selective capture of lead [36] from wastewater. The presence of MOF in SWCNT BPs was found to be beneficial for the adsorption of all tested lanthanides with the Ce3+ recovery reaching the value of 263.30 mg of cerium adsorbed per gram of MOF-SWCNT BP. This was related to the alcohol functionalities of threonine, within MOF pores. Similarly, the lead adsorption capacity of neat BPs increased by 42% when 25 wt. % SWCNTs were substituted by MOFs. Moreover, stable BPs for the adsorption of heavy metal ions were obtained by the noncovalent interaction of oxidized CNTs with GO [51]. The presence of nanotubes stabilized the mechanical properties of BPs, but reduced the adsorption capacities for all tested heavy metals. In particular, the experimental lead adsorption capacity by BPs with GO content up to 90 wt. % was lower than the experimental maximum loading capacity by neat CNT BPs (163 ± 10 mg g-1), due to the presence of micro- and nano-channels among entangled CNTs.”
[46] Huang, J.-Q.; Xu, Z.-L.; Abouali, S.; Garakani, M.A.; Kim, J.-K. Porous graphene oxide/carbon nanotube hybrid films as interlayer for lithium-sulfur batteries. Carbon 2016, 99, 624-632. DOI: 10.1016/j.carbon.2015.12.081.
[47] Yuan, G.-j.; Xie, J.-F.; Li, H.-H.; Shan, B.; Zhang, X.-X.; Liu, J.; Li, L.; Tian, Y.-Z. Thermally reduced graphene oxide/carbon nanotube composite films for thermal packaging applications. Materials 2020, 13, 317. DOI: 10.3390/ma13020317.
[48] Minett, A.; Fràysse, J.; Gang, G.; Kim, G.-T.; Roth, S. Nanotube actuators for nanomechanics, Curr. Appl. Phys. 2002, 2, 61-64 DOI: 10.1016/S1567-1739(01)00100-6.
[49] Kausar, A.; Ilyas, H.; Siddiq, M. Current research status and application of polymer/carbon nanofiller buckypaper: a review. Polym. Plast. Technol. Eng. 2017, 56, 1780-1800, DOI: 10.1080/03602559.2017.1289407.
[50] Alshahrani, A.A.; Al-Zoubi, H.; Nghiem, L.D.; in het Panhuis, M. Synthesis and characterisation of MWNT/chitosan and MWNT/chitosan-crosslinked buckypaper membranes for desalination, Desalination 2017, 418, 60-70. DOI: 10.1016/j.desal.2017.05.031.
[51] Musielak, M.; Gagor, A.; Zawisza, B.; Talik, E.; Sitko, R. Graphene oxide/carbon nanotube membranes for highly efficient removal of metal ions from water, ACS Appl. Mater. Interfaces 2019, 11, 28582-28590. DOI: 10.1021/acsami.9b11214.
- It is also mentioned that the use of GO-SWCNT BPs could be a cheap and effective way for water decontamination from lead. It remains insufficiently justified in the manuscript. Comparison with other methods and materials in terms of economic price and technological efficiency should be done.
- We thank the reviewer for the important raised question, regarding the perspective on industrial feasibility and scale up of the proposed technology. At this stage it is more difficult to estimate the operating costs of the Pb2+ removal (in €/g of Pb2+) by GO-SWCNT BPs and compare these costs with other existing technologies. Nevertheless, according to a previously reported estimation for the production of BPs (hosting MOFs at 25 wt.% for a 15 kg scale), the costs of BPs and MOFs were estimated in around 1000 €/kg and less than 4000 €/kg, respectively. Considering that the GO production costs are noticeably lower than those of MOFs, it is possible to assess that the production of GO-SWCNT BPs is of course cheaper than that of MOF-SWCNT BPs or pure-MOF based devices. The following sentence was added (lines 242-244): “Regarding the industrial feasibility and scale up, at this stage it is difficult to estimate the operating costs for the lead removal by the proposed technology and compare them with those of other existing technologies. According to a previously reported estimation for the production of BPs (hosting MOFs at 25 wt.% for a 15 kg scale), the costs of BPs and MOFs were estimated in around 1000 €/kg and less than 4000 €/kg, respectively [36]. Considering that the GO production costs are noticeably lower than those of MOFs [57], it is possible to assess that the production of GO-SWCNT BPs is cheaper than that of MOF-SWCNT BPs or pure-MOF based devices.”
[57] Ranjan, P.; Agrawal, S.; Sinha, A.; Rao, T.R.; Balakrishnan, J.; Thakur, A.D. Low-cost non-explosive synthesis of graphene oxide for scalable applications. Sci. Rep. 2018, 8, 12007. DOI: 10.1038/s41598-018-30613-4.

Reviewer 2 Report
Comments from Reviewer
Title: GO-SWCNT buckypapers as an enhanced technology for water decontamination from lead
The current form's presentation of methods and scientific results is unsatisfactory for publication in the Molecules journal. The minor drawbacks to be addressed can be specified as follows:
1. The porosity and/or the chemical nature of materials are important factors influencing the adsorption capacity. Such information is missing. The lack of this research disqualifies this work. I am interested in how the porosity changed with the addition of GOs. I am interested in how the porosity has changed as the GOs are added. Similarly, the chemical nature of the materials tested is changing.
2. Line 56. buckypapers (BP) ---> buckypapers (BPa). See line 19: SWCNT buckypapers (SWCNT BPs).
3. Lines 64-66. What are the goals of the paper?
4. Line 206. There is no information about SWCNTs. Please provide details. Has this material been used before? Are these commercial materials? Please also provide appropriate references if necessary. GOs?
5. Line 209. COOH-SWCNT? What is this material? How do you know if it has such functional groups?
6. Line 70. PTFE?
7. Line 70, average pore size 0.45 μm. How do the authors know this?
8. Fig. 3, x and y axes. Units in parentheses (). See Fig. 4.
9. Fig. 4. Did the authors try to produce Go BPs or 99% GO-SWCNT BPs or 95% GO-SWCNT BPs?
Sincerely,
The reviewer.
Author Response
Reply to the referees’ comments (Ms # molecules-1771507)
We thank the referees for their valuable comments and suggestions, which helped us much in the revision of the manuscript. In the following pages, we report the responses to their comments.
Reviewers' Comments to Authors
Reviewer # 2
- The porosity and/or the chemical nature of materials are important factors influencing the adsorption capacity. Such information is missing. The lack of this research disqualifies this work. I am interested in how the porosity changed with the addition of GOs. I am interested in how the porosity has changed as the GOs are added. Similarly, the chemical nature of the materials tested is changing.
- We agree with the comment of reviewer # 2. The porosity values of BPs were evaluated as described in 3.2. Characterization procedure and reported in Table 1. The chemical nature of used materials, present in §3.1. Buckypaper preparation, was enriched with info obtained from the datasheets of the supplier (see also reply to comments #4).
The following sentences were added:
“The percentage porosity of BP, P(%), was estimated by measuring the weight of a wetting liquid (3M-FC-40, 3M Italia Srl, Pioltello, Milan, Italy), contained in the membrane pores at 25 °C, according to the following Equation (4):
(4)
where Ww and Wd are the weight of the wet and dry samples, dw and dm the density of wetting liquid (1.855 g cm-3) and BP membranes, respectively.”
“The adsorbent physico-chemical properties, such as water wettability, surface area, and chemical nature, can affect the adsorption of heavy metals in addition to the operating conditions (e.g. adsorbent amount, temperature, pH and ionic strength values, adsorption time, initial concentration, and interferent presence). The presence of surface functional groups, such as hydroxyl and carboxyl, can improve wettability and heavy metal chelation. Therefore, the increased lead adsorption capacity shown by GO-SWCNT BPs with larger amounts of GO (from 42.18% for SWNT BPs to 59.95% for 75% GO-SWCNT BP) can be attributed to the beneficial presence of functional groups able to enhance membrane wettability and metal chelation and overcome the possible drawbacks due to the reduced porosity (see Table 1).”
“On the contrary, the porosity percentage found for BPs decreased for larger GO amount as a consequence of a closer packing between GO sheets.”
- Line 56. buckypapers (BP) ---> buckypapers (BPs). See line 19: SWCNT buckypapers (SWCNT BPs).
- Thank you for the suggestion; done.
- Lines 64-66. What are the goals of the paper?
- Thank you for the suggestion: the sentence in lines 64-66 (from lines 81 in the revised manuceript) was changed as follows: “In this paper, single walled carbon nanotube/graphene oxide buckypapers (GO-SWCNT BPs) were prepared and characterized as novel membranes for the adsorption of lead from wastewater. The adsorption capacity and selectivity towards Pb2+ (both from neat solutions and in the presence of other interferents) by GO-SWCNT BPs were evaluated as a function of the increasing percentage (wt.) of GO used in their preparation. Hybrid membranes showed high lead adsorption capacity and interesting removal properties also towards other heavy metal ions, allowing the possibility of a multielement decontamination of wastewater.”
- Line 206. There is no information about SWCNTs. Please provide details. Has this material been used before? Are these commercial materials? Please also provide appropriate references if necessary. GOs?
Line 209. COOH-SWCNT? What is this material? How do you know if it has such functional groups?
Line 70. PTFE?
Line 70, average pore size 0.45 μm. How do the authors know this?
- We apologize for the missing information and unexplained acronyms. We explained both in the Section 2. Results and1. Buckypaper preparation:
- the used acronyms,
- the commercial nature of materials,
- and the origin of added properties (data sheet from suppliers).
- Fig. 3, x and y axes. Units in parentheses (). See Fig. 4.
- We apologize for the misprints. Units of x and y axes are now in parentheses.
- 4. Did the authors try to produce Go BPs or 99% GO-SWCNT BPs or 95% GO-SWCNT BPs?
As reported in the caption of Figure 1 and in Lines 75-77 of the previous version, the substitution of SWCNT mixture with GO amounts larger than 75 wt. % gave cracked films, as reported in Figure 1 (e) for 85% GO-SWCNT BPs, with difficult detachability and, consequently, broken membranes. In order to stress this drwback, the following sentence was added in the Section §3.1: “The substitution of SWCNT mixture with GO amounts larger than 75 wt. % gave cracked films, as reported in Figure 1 (e) for 85% GO-SWCNT BPs, with difficult detachability and, consequently, broken membranes. Therefore, the maximum GO loading in GO-SWCNT BPs was set in 75 wt.%.”

Round 2
Reviewer 2 Report
The authors have made the essential corrections, provided some detailed answers to some of the questions, and ignored some of the comments. Overall the manuscript improved. In the future, I suggest that the authors focus more attention on researching the chemical nature of the surface.